# The Impact of a Healthy Lifestyle on Lower Urinary Tract Symptoms and Erectile Function: A Prospective Study

**DOI:** 10.3390/healthcare13020185

**Published:** 2025-01-18

**Authors:** Riccardo Lombardo, Matteo Romagnoli, Chiara Splendore, Luca Sarcinelli, Beatrice Turchi, Giacomo Gallo, Giorgia Tema, Antonio Franco, Antonio Nacchia, Ferdinando Fusco, Yazan Al Salhi, Andrea Fuschi, Antonio Pastore, Antonio Cicione, Andrea Tubaro, Cosimo De Nunzio

**Affiliations:** Department of Urology, Sapienza University of Rome, 00185 Rome, Italy; matteo.romagnoli@uniroma1.it (M.R.); chiara.splendore@uniroma1.it (C.S.); lucasarcinelli94@gmail.com (L.S.); beatrice.turchi@uniroma1.it (B.T.); giacomo.gallo@uniroma1.it (G.G.); giorgiat88@hotmail.it (G.T.); antonio.franco@uniroma1.it (A.F.); antonio.nacchia@uniroma1.it (A.N.); ferdinando-fusco@libero.it (F.F.); yazan.alsalhi@uniroma1.it (Y.A.S.); andrea.fuschi@uniroma1.it (A.F.); antopast@hotmail.com (A.P.); acicione@libero.it (A.C.); andrea.tubaro@uniroma1.it (A.T.); cosimo.denunzio@uniroma1.it (C.D.N.)

**Keywords:** LUTS, lifestyle, young, smoking, cannabis

## Abstract

**Background and Objectives**: The impact of lifestyle on lower urinary tract symptoms has been deeply evaluated in recent years; however, studies in the young population are missing. The aim of this study is to evaluate the impact of alcohol intake, tobacco and cannabinoid smoking, physical activity, and dietary regime on urinary symptoms and sexual function in young adults under 30 years of age. **Methods**: A prospectively enrolled population of healthy young adults of both sexes under 30 years of age was selected. Young people with comorbidities were excluded. All participants were assessed by completing an anonymous questionnaire which included questions on medical history, lifestyle, smoking and alcohol intake, urinary symptoms, and sexual function only in male subjects. The questionnaire was postponed in case there was an acute pathology. **Results**: Overall, 802 young adults were prospectively enrolled, of whom 44% were male and 56% female, with a median age of 26 (23/28) years. In our population, 580/818 (70.9%) subjects presented an IPSS ≥ 3. In the analysis of the association between urinary symptoms and smoking, smokers presented urinary symptoms more frequently than nonsmokers (76% vs. 61%; *p* < 0.05). No association between urinary symptoms and alcohol intake, cannabinoid smoking, physical activity, and dietary regimen was recorded. On multivariable analysis, smokers had an almost doubled risk of urinary symptoms compared to nonsmokers (OR: 1.78; *p* = 0.001). **Conclusions**: In conclusion, we demonstrated how even in the young population there can be a correlation between LUTSs and different lifestyles.

## 1. Introduction

Lower urinary tract symptoms (LUTSs) are a common complaint in the aging male population. LUTSs affect 8% of men between 31 and 40 years old, with its prevalence reaching 90% in men above 90 years old.

In women, the prevalence of LUTSs increases with aging too; in women older than 70 years old, more than 40% of them are affected by LUTSs [1].

LUTSs can be distinguished in voiding, storage, and post-micturition symptoms and are often considered a consequence of benign prostatic hyperplasia (BPH). BPH is a common cause of bladder outlet obstruction (BOO) with an increasing incidence in men over 50 years of age. Bladder dysfunctions may also cause BOO, like detrusor overactivity (OAB) and detrusor underactivity (DU); other dysfunctions of the urinary tract can be causes of LUTSs [2]. Persistent prostatic inflammation can develop and may contribute to the development and progression of BPH [3]. An inflammatory reaction in prostatic tissue can be triggered by several factors, including bacterial infections, viruses (e.g., human papilloma virus, herpes simplex virus type 2, and cytomegalovirus), sexually transmitted organisms (e.g., gonorrhoeae and chlamydia), hormones, metabolic syndrome, dietary factors, and urinary reflux, as well as an autoimmune response [4,5]. The infiltrating inflammatory cells (70–80% T-lymphocytes, 10–15% B-lymphocytes, and 15% macrophages) become activated and release pro-inflammatory cytokines, which in turn increase the expression of several growth factors (e.g., interleukin (IL)-17, IL-15, IL-8, interferon-γ, fibroblast growth factor (FGF), and FGF-2), resulting in an abnormal proliferation of epithelial and stromal cells. The subsequent increased oxygen demand of these cells leads to local hypoxia, producing low levels of reactive oxygen species (ROS) and promoting angiogenesis and the production of additional growth factors (i.e., vascular endothelial growth factor, IL-8, FGF-2, FGF-7, and transforming growth factor ß) [4]. As such, persistent prostatic inflammation or chronic prostatic inflammation is a histological observation and, irrespective of the mechanism that triggers the uncontrolled inflammatory response, the final result of this process induces tissue damage with subsequent abnormal wound healing and stromal and epithelial cell proliferation, and thus BPH.

The overall prevalence of moderate-to-severe LUTSs in the young adult population is 16% but increases rapidly with age [6].

About half of young adults (aged 39 and younger) may be affected by a variety of conditions that can cause LUTSs. Filling-phase symptoms are almost two-fold more common than emptying-phase symptoms. Infectious or inflammatory causes are the most frequent causes of LUTSs in this age group, followed by LUTSs due to other etiologies that include urethral strictures and primary bladder neck obstruction, as well as neurogenic and non-neurogenic bladder dysfunction [7]. In addition, in young adults, the presence of a bladder neck stricture is a common condition that is often underestimated. Its incidence in male patients with LUTSs between 18 and 50 years old is 28–54% according to epidemiological studies. These patients usually present a low body mass index, low PSA levels, severe urinary symptoms, and poor flow [8,9,10]. Recently, the prevailing theory on the pathophysiology of this condition has been challenged, moving from a congenital condition to an acquired inflammatory condition [11]. More specifically, an initial infection causes a chronic inflammatory condition, which then leads to bladder neck sclerosis. Future studies should better assess the etiology of this condition in young adults. Regarding neurological conditions, several congenital and acquired conditions may cause a neurogenic bladder, leading to different urodynamic profiles, which are not within the scope of this article.

Persistent LUTSs can present in younger women under 40 years of age with an unclear etiology, which may be characterized using video urodynamic studies. The most common etiology found in this kind of patient is dysfunctional voiding followed by detrusor overactivity [12].

The best-known risk factor for both sexes is increasing age.

Furthermore, studies conducted on populations over the age of 40 have demonstrated how lifestyle (including obesity, physical activity, alcohol, smoking, diet) represents an important modifiable risk factor for LUTSs in both sexes [13].

These lifestyle variables offer possible therapeutic targets to delay disease onset, prevent progression, or attenuate symptoms.

In particular, low levels of physical activity and a high body mass index (BMI) are well-documented risk factors for LUTSs in both men and women [14]. The study conducted by Raheem O. and Parsons J. highlights that regular, medium-to-high-intensity physical exercise can significantly lower the risks of BPH and LUTSs [15]. Both studies found that engaging in moderate-to-vigorous physical activity was linked to up to a 25% reduction in the risk of BPH or LUTSs, with the protective benefits increasing as activity levels rose.

Smoking research has shown that regular smokers are at an increased risk of LUTSs (OR 2.01; 95% CI 1.04–3.89; *p* < 0.05) compared to former smokers or nonsmokers [16]. In particular, smokers have a urination frequency and urgency that are more than three times higher than normal [17]. Smoking can also cause hormonal and nutritional imbalances that affect the bladder and has anti-estrogenic effects in women [18].

High-calorie diets, starches, and red meat appear to be weakly associated with an increased risk of benign prostatic hyperplasia (BPH), the most frequent cause of LUTSs in men over 40, while a diet rich in vegetables (OR = 0.66; *p* < 0.05) is associated with a low risk of BPH in men and LUTSs in both sexes [19].

The results of studies on micronutrients highlight how carotene (OR = 0.80; *p* < 0.05) and lycopene are implicated in decreasing the risk of BPH, while zinc intake (OR = 1.10; *p* < 0.05) and sodium (OR = 1.30; *p* < 0.05) may increase the risk [20].

Furthermore, overall poor nutrition and limited dietary variety appear to be associated with clinically significant LUTSs in both sexes.

Protein intake can reduce the risk of BPH [18] but on the contrary appears to increase the risk of symptoms of the filling phase in women [21]. Finally, in elderly subjects, it was highlighted that the intake of isoflavone (a phytoestrogen contained in numerous legumes and soy) has a protective role against LUTSs (OR 0.59; 95% CI 0.44, 0.80; *p* < 0.05) [22].

Numerous systemic diseases in men between the ages of 40 and 70 represent risk factors for BPH but also for erectile dysfunction. Recent data demonstrate that diabetes is the main systemic disease with a three times higher risk of developing BPH and ED [17]. Cardiovascular disease, hypertension, high lipid levels [16], and chronic alcoholism have also been found to be risk factors for BPH and erectile dysfunction [17].

Data on alcohol are somewhat controversial. A moderate alcohol intake appears to be protective against BPH, but the same effect does not appear to apply to LUTSs in both sexes [13]. Several systematic reviews have examined the link between alcohol consumption and urinary symptoms, often with conflicting results. Rohrmann S. found that consuming ≥38 g/day of alcohol reduced the risk of LUTSs (OR = 0.41; *p* < 0.05), while small daily doses also lowered the risk (OR = 0.59; *p* < 0.05) [16]. Similarly, reviews by Bradley C. and Platz E. reported that a moderate alcohol intake (30.1–50 g/day) was associated with a reduced risk of BPH and LUTSs (OR = 0.59; *p* < 0.05), though the protective effect diminished at higher levels (≥50.1 g/day, OR = 0.72; *p* < 0.05) [16,23].

Comorbid conditions specific to women were also examined as risk factors for LUTSs.

Women who had a vaginal birth were more likely (RR 5.8, 95% CI 1.2–33.7; *p* < 0.05) to report stress urinary incontinence (SUI) than those who gave birth by cesarean section, in particular a year or more after giving birth [18,24].

Having a hysterectomy has also been shown to increase the likelihood of LUTSs, including hesitancy, incomplete voiding, post-void dribbling, frequency, urgency, and urge incontinence [13]. Postmenopausal status and a longer duration of perimenopause have been associated with filling and emptying symptoms, including stress urinary incontinence, nocturia, and weak flow [13].

Another risk factor of recent scientific interest is the intake of cannabinoids; there are still few data, but from some of the research, it emerges that moderate daily use may have a protective role against LUTSs [25,26].

Finally, all these factors can vary greatly based on region, culture, occupation, and socio-economic status.

In the past few years, several studies have evaluated the role of lifestyle habits and physical activity in patients with LUTSs and BPH, with conflicting results. Most of these studies include male patients over 50 years old, excluding younger patients. Therefore, very little is known about the association between LUTSs and lifestyle in the young population below 30 years old.

### Objective and Hypothesis

The aim of the present study is to evaluate the association between diet, physical activity, smoking, alcohol consumption, and LUTSs in young adults. The hypothesis is that an adequate lifestyle is associated with a lower risk of LUTSs in this population of young adults.

## 2. Materials and Methods

### 2.1. Participants and Data Collection

A prospectively enrolled population of healthy young adults under 30 years of age was selected. Male and female subjects were included and young people with comorbidities were excluded from the analysis. More precisely, all subjects with hereditary diseases and chronic diseases were excluded. The decision to exclude patients with comorbidities was made to homogenize the enrolled population and exclude possible biases in the analysis.

All participants were assessed by having them complete an anonymous questionnaire, which included questions on medical history (both clinical and pharmacological), lifestyle, smoking and alcohol intake [27], urinary symptoms, and sexual function (only in male subjects).

The questionnaires were postponed in case there was an acute disease.

The data collected in the medical history included age, sex, weight, height, occupation, and any pathologies and medications (in particular 5-alpha reductases and alpha-1 blockers).

### 2.2. Instruments

Alcohol and smoking (tobacco and cannabis) intake was assessed using specific questionnaires that assessed their frequency and quantity.

Physical activity was assessed with the GPAQ questionnaire [28], which measures the type and quantity of physical activity that the subject normally performs. The questions submitted refer to the activity carried out in the last 7 days at work, on daily travel, and in free time.

The dietary score was obtained through the Mini ECCA questionnaire [29], which investigates the consumption of fruit, vegetables, types of meat, sugary drinks, legumes, cereals, and meals outside the home.

The urinary symptom evaluation was evaluated by the IPSS and the International Consultation on Incontinence Questionnaire Overactive Bladder Module (ICIQ-OAB) questionnaires.

Sexual function was determined by administering the short version of the International Index of Erectile Function (IIEF) erectile function questionnaire to male participants only.

### 2.3. Data Analysis

Statistical analysis was performed using SPSS 27.0. Continuous variables were described using means and standard deviations and medians with interquartile ranges. Categorical variables were described as fractions and percentages. Statistically significant differences were assessed with the Mann–Whitney test and the chi-square test for continuous and categorical variables, respectively.

Univariate and multivariable binary logistic regression were used to evaluate risk factors for urinary and sexual dysfunction.

The sample size was calculated using statistical software, estimating an 8% prevalence of LUTSs in this population. The hypothesis was that individuals with LUTSs exhibit poor lifestyle choices twice as frequently as those without LUTSs. We used a 95% confidence interval and 90% power. The minimum sample size required was 680 patients. Considering a 20% ineligibility rate, the adjusted sample size was 816.

### 2.4. Bioethical Committee Approval

This study was evaluated and approved by the Bioethical Committee of our University (Prot. 258 SA_2021 RIF. CE 6376_2021—Studio Clinico: “IRU STUDY”).

## 3. Results

Overall, 816 young adults were enrolled and 802 were eligible for analysis. In total, 44% were male and 56% were female, with a median age of 26 (23/28) years. The female subjects were younger (median: 25 vs. 27, *p* < 0.05) with a lower BMI (median: 22 vs. 25, *p* < 0.05) and had a greater IPSS (median: 5 vs. 4, *p* < 0.05) and a greater number of nocturia episodes (median: 1 vs. 0, *p* < 0.05) compared to the male subjects (Table 1).

A total of 569/802 (71%) smoked tobacco, with an average number of cigarettes in 24 h of 5 (1/10). In the comparison between men and women, no statistically significant differences were highlighted in terms of frequency, units, or time of smoking (Table 2).

A total of 722/802 (90%) drank alcohol, with a median number of alcohol units of 2 (1/4). When comparing men and women, men consumed alcohol more frequently and for a longer time. The data are reported in Table 3.

Overall, 441/802 (55%) people smoked cannabis, with a median THC unit intake of 1 (0/2). When comparing men and women, men consumed THC more frequently and for a longer time. The data are reported in Table 4.

The calculated median physical activity was 2100 MET (4080/818). A total of 44% of the sample performed intense physical activity. When comparing men and women, men performed more intense physical activity in terms of MET. The data are reported in Table 5.

The median diet score was 8 (6/9). The subjects who followed a healthy diet constituted 22% of the sample. When comparing men and women, women had a healthier diet than men. The data are reported in Table 6.

For the assessment of urinary symptoms: the median IPSS was 4 (2/8) and the median total OAB score was 22 (20/26). The evaluation of sexual function showed a median IIEF of 25 (24/25).

In our population, 580/818 (70.9%) subjects presented an IPSS ≥ 3. There were no statistically significant differences in terms of age or sex between subjects with and without urinary symptoms (Table 7).

In the analysis of the association between urinary symptoms and smoking, smokers presented urinary symptoms more frequently than nonsmokers (76% vs. 61%; *p* < 0.05). There were no statistically significant differences in terms of the number of cigarettes, frequency, or time of smoking between patients with and without urinary symptoms (Table 8).

In the analysis of the association between urinary symptoms and alcohol, alcohol consumers presented urinary symptoms more frequently than abstainers (95% vs. 86%; *p* < 0.05). There were no statistically significant differences in terms of the number of units of alcohol, frequency, and time of intake between subjects with and without urinary symptoms (Table 9).

In the analysis of the association between urinary symptoms and THC consumption, THC users did not present urinary symptoms more frequently than non-users (*p* > 0.05). There were no statistically significant differences in terms of the number of THC units, frequency, or time of intake between subjects with and without urinary symptoms (Table 10).

In the analysis of the association between urinary symptoms and physical activity, it was found that the median number of METs was comparable between subjects with and without urinary symptoms (2160 vs. 1815; *p* = 0.204). Furthermore, there were no statistically significant associations between the level of physical activity and urinary symptoms (Table 11).

In the analysis of the association between urinary symptoms and diet, subjects with urinary symptoms did not have a statistically different Mini ECCA score compared to subjects without urinary symptoms. Furthermore, the total IPSS was not correlated with the Mini ECCA value (rho = 0.005; *p* = 0.899). Subjects who did not eat processed cereals, did not eat out, and did not eat red meat more frequently had IPSS ≥ 3 (Table 12).

In the age-adjusted logistic regression analysis reported in Table 13, smokers had almost double the risk of urinary symptoms compared to nonsmokers (OR: 1.78; *p* = 0.001). In our univariate analysis, alcohol and cannabinoid consumption, diet, and physical activity did not represent risk factors for the presence of urinary symptoms, and for this reason, they were not evaluated as possible risk factors in the multivariate analysis.

## 4. Discussion

For this study, we prospectively selected a sample of young adults between 18 and 30 years old. Our study has the merit of analyzing a homogeneous population of young adults coming from the same geographical region, thus minimizing possible biases related to genetics and comorbidities. Furthermore, subjects were prospectively enrolled and systematically assessed with internationally validated questionnaires. Our study comprehensively highlights how some lifestyles such as cigarette smoking represent a risk factor for LUTSs. On the other hand, it highlights that in a population of young adults, the impact of diet, drugs, and alcohol is not significant on the presence of LUTSs.

Although urinary symptoms are an inevitable consequence of aging, further findings from this study were associated with several cofactors such as comorbidities and different lifestyles. There are numerous studies that have focused on possible pathophysiological mechanisms underlying the development of LUTSs. A central and fundamental role is played by inflammation. A sedentary lifestyle, poor physical activity, cigarette smoking, alcohol abuse, depression, hypertension, cardiovascular diseases, hyperlipidemia, diabetes mellitus, obesity, high BMI, and hypogonadism are all potential factors triggering prostate and/or bladder inflammation.

De Nunzio and collaborators [3] analyzed these possibilities in a systematic review on the pathophysiological mechanism of LUTSs. The cells of the inflammatory infiltrate such as lymphocytes and macrophages are activated by numerous growth factors, such as IL8, FGF-2, FGF-7, and TGF-beta. Any type of stimulus triggers the activation of a cascade mechanism, resulting in hypoxia and the production of free radicals and reactive oxygen species (ROS). This, therefore, results in cellular damage and consequent abnormal reparative tissue proliferation of the cells under the stimulus of these growth factors (Figure 1). Similarly, in the bladder, reduced oxygenation/ischemia at the pelvic level can lead to an alteration of the urothelium and bladder afferent endings, triggering the inflammatory cascade [3].

In the literature, several systematic reviews have reported studies on the relationship between alcohol consumption and the risk of developing urinary symptoms, with sometimes conflicting results. The study by Rohrmann S. highlighted how an intake of ≥38 g of alcohol per day was associated with a reduced risk of LUTSs (OR = 0.41, 95% CI 0.14–1.2; *p* < 0.05) compared to subjects who did not ever take any [16]. Men who consumed alcohol daily but in small doses reported a lower risk of developing LUTSs (OR = 0.59; 0.37–0.95; *p* < 0.05). Similarly, systematic literature reviews by Bradley C. and Platz E. reported that moderate alcohol consumption is associated with a reduced risk of BPH and LUTSs (30.1–50 g/day vs. 0: (OR) = 0.59, 95% CI 0.51–0.70; *p* < 0.05), although the relationship was attenuated for high levels (greater than or equal to 50.1 g/day vs. 0: OR = 0.72, 95% CI 0.57–0.90; *p* < 0.05) [23,30].

Finally, a further study suggests that habitual alcohol consumers are less likely to have LUTSs than nondrinkers (OR = 0.59; 95% CI 0.37–0.95; *p* = 0.07), but an intake of more than 72 g/day (approximately five or more drinks per day) is associated with a higher probability of reporting LUTSs compared to nondrinkers [31].

Our results are in line with the aforementioned Norwegian study confirming that 95% of young adults do not report urinary symptoms associated with alcohol [6].

In recent years, numerous studies have been published on the association between tobacco smoking and LUTSs, with conflicting results [23].

Data regarding nocturia associated with smoking as a risk factor for LUTSs were conflicting in both male and female subjects, depending on the samples examined. In one study, no particular associations were found between smoking and LUTSs (*p* < 0.05) [32]. On the contrary, other studies suggest a correlation between smoking and increased frequency and urgency of urination in women, while in men, no association between LUTSs and tobacco has been found. In particular, the consumption of 16 or more cigarettes per day compared to 0 cigarettes per day was evaluated in the Asplund study (OR, 1.8; 95% CI, 1.1–2.8; *p* < 0.05) and in Tettamanti’s study (OR 1.5, 95% CI 1.18–1.89) [33,34].

In the systematic review by Rohrmann S., no study demonstrated a statistically significant association between cigarette smoking and LUTSs (OR= 2.01; 95% CI 1.04–3.89; *p* < 0.05) [16], while in the Platz study, a significant association appeared between smoking and LUTSs; in fact, heavy smokers reported a greater risk of developing urinary symptoms compared to nonsmokers (OR = 1.45, 95% CI 1.07–1.97; *p* <0.05), and moderate smokers did not [30]. Our data form a positive correlation with the Platz E study, confirming an almost doubled risk of developing LUTSs in habitual smokers compared to nonsmokers. The present study adds further evidence on the role of cigarette smoking, particularly in young adults.

Regarding smoking cannabis, it is not to be considered a risk factor for the development of urinary symptoms according to the results of our study and other experiences with other populations. On the other hand, all these studies highlight a possible benefit and possible therapeutic use of cannabis with low doses of THC (OR = 0.55; 95% CI 0.408–0.751, *p* > 0.01) for the treatment of LUTSs [27,35].

Our study did not report statistically significant data regarding physical activity and diet, but several studies support weight loss and physical exercise in the context of standard treatments for symptomatic LUTSs. In fact, the studies by Raheem O. and Parsons J. reported that medium–high and constant physical exercise can reduce the risks of BPH and LUTSs. Both studies concluded that moderate-to-vigorous physical activity was associated with up to a 25% reduction in the risk of BPH or LUTSs, with a protective effect increasing with higher levels of activity [14,15]. From the studies by Bradley C. and Kristal et al., it was found that a high-calorie diet rich in starches and red meat is weakly associated with BPH, while a further lower risk is associated with a diet rich in vegetables and polyunsaturated fats with low saturated fat content [23,36]. Micronutrient studies identified carotene (OR = 0.80; *p* < 0.05) as a protective factor against LUTSs and BPH, while zinc (OR = 1.10; *p* < 0.05) could increase its risk [20]. Regarding urinary incontinence, it was reported by different studies how a low-calorie diet decreased UI in prediabetic women and how a modest weight loss of between 5% and 10% improved symptoms and reduced the number of UI episodes by 70% [23,37].

Clinically significant LUTSs have been linked with a poorly varied diet and sodium intake in men and increased total calories in women [38,39].

Again, in these same studies, it was found that protein intake can reduce the risk in men but increases bloating symptoms in women.

In our study, the consumption of red meat was associated with a lower risk of lower urinary tract symptoms, which is in line with the data from the aforementioned study.

In the present study, no clear association was observed between alcohol, diet, physical activity, and LUTSs defined as IPSS ≥ 3. The negative results clearly depend on the enrolled population. Overall, most of the enrolled population (>80%) presented moderate consumption of alcohol and performed moderate physical activity, and only 3% had a diet that posed a risk to their health. In addition, the present study did not evaluate other possible factors that may influence LUTSs, such as sleeping patterns, stress, workload, and working shifts [40,41,42,43]. Recently, Chen et al. found in their cross-sectional study that sleeping less than 6 h a night increases the risk of LUTSs (OR: 1.38; *p* < 0.05). The present study confirms that, in a young population, a slight variation in lifestyle does not have an impact on urinary symptoms. Future studies in different populations, such as young obese individuals, young alcoholic individuals, or severely sedentary young adults, may yield different results.

It has been reported in the literature that protein intake is associated with better sexual function. It is estimated that there will be approximately 322 million men affected by erectile dysfunction (ED) by 2025. It is reasonable that our study did not reveal statistically significant data regarding sexual function and lifestyles since, as with BPH, the prevalence of ED increases with age and other risk factors are cardiovascular disease, physical inactivity, obesity, metabolic syndrome, and diet [44]. In our cohort, the prevalence of ED was too low to perform an in-depth analysis on it.

Our study has some limitations. Our analysis applies to a series of subjects under 30 years of age and cannot be extended to other populations with different characteristics, lifestyle habits, and regional or cultural factors. Despite the large sample size, the low incidence of erectile dysfunction and of serious urinary symptoms limits the analysis. Nevertheless, a study on a larger scale is ongoing and the results will soon be available. Finally, our study evaluated lifestyles at a specific point in time and could not evaluate changes in lifestyles over time. In this regard, a prospective study is underway to evaluate the effect of lifestyle changes on urinary symptoms. Finally, the present study presents the common characteristics of a cross-sectional study, and therefore, studies with a different design and longer follow-up are ongoing. Despite the limitations, this represents one of the few papers present in the literature on the relationship between LUTSs and lifestyles in the young population (<30 years old).

## 5. Conclusions

In conclusion, the present study suggests a correlation between LUTSs and different lifestyles in the young population. In particular, cigarette smoking was the most represented risk factor for urinary symptoms in the under-30 population, with an almost doubled risk for habitual smokers compared to nonsmokers. The role of cannabis in the clinical picture of LUTSs still remains to be defined. Diet, physical activity, and alcohol intake showed no statistically significant influence in young adults. Nevertheless, given the extensive literature on samples of adult subjects, a healthy diet with a reduced alcohol intake and moderate-to-high physical activity should be recommended. These data also highlight how prevention and awareness campaigns on certain voluptuary habits and on quitting smoking can have a significant impact on urinary health, even among the young population. The present study presents some limitations related to the cross-sectional design and characteristics of the enrolled population. Future studies, with different designs and a longer follow-up period, may improve our knowledge of the complex association between lifestyles and LUTSs. In particular, the impact of lifestyle changes on LUTSs should be evaluated.

## Figures and Tables

**Figure 1 healthcare-13-00185-f001:**
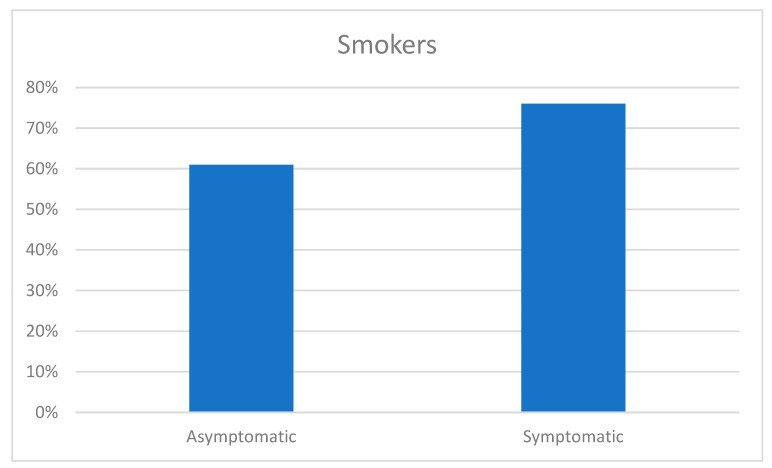
Associaation between smoking and IPSS ≥ 3.

**Table 1 healthcare-13-00185-t001:** Population characteristics.

Subjects	Total 802	Men 352 (44%)	Women 450 (56%)	*p*
Age (years)	26 (23/28); 26.9 ± 6.7	27 (24/29)	25 (23/28)	0.001
BMI (Kg/m^2^)	24 (22/26); 24.5 ± 3.2	25 (23/27)	22 (20/24)	0.001
IPSS	4 (2/8); 3 ± 1	4 (2/6)	5 (2/8)	0.001
Nocturia	1 (0/1); 0.8 ± 1	0 (0/0)	1 (0/1)	0.001
OAB score	22 (20/26); 24.4 ± 8	21 (19/22)	22 (20/23)	0.325
IIEF	4 (4/5); 4.2 ± 0.7	n.a.	n.a.	

**Table 2 healthcare-13-00185-t002:** Cigarette smoking characteristics.

	Total	Men 352	Women 450	*p*
Tobacco smokers	569/802 71%	253/352 72%	319/450 71%	0.817
Smoke time (<5 years)	187/569 30%	66/253 26%	105/319 33%	0.270
Smoke time (5–10 years)	293/569 47%	124/253 49%	147/319 46%	
Smoke time (>10 years)	143.5/569 23%	63/253 25%	67/319 21%	
Tobacco units (number of cigarettes)	5(1/10); 6.5 ± 6	5 (2/12)	4 (1/10)	0.172
Smoke frequency (1–2 days a month)	100/569 16%	35/253 14%	54/319 17%	0.347
Smoke frequency (3–9 days a month)	87/569 14%	38/253 15%	41/319 13%	
Smoke frequency (10–19 days a month)	62.4/569 10%	17.5/253 7%	41/319 13%	
Smoke frequency (>20 days a month)	374.4/569 60%	157/253 62%	182/319 57%	

**Table 3 healthcare-13-00185-t003:** Alcohol consumption characteristics.

	Total	Men 317	Women 405	*p*
Alcohol consumption	722/802 90%	298/317 94%	356/405 88%	0.010
Alcohol time (<5 years)	174/722 22%	45/298 15%	99/356 28%	0.001
Alcohol Time (5–10 years)	380/722 48%	134/298 45%	182/356 51%	
Alcohol time (>10 years)	237/722 30%	119/298 40%	75/356 21%	
Alcohol units	2 (1/4); 8 ± 54	3 (2/5)	2 (1/4)	0.001
Alcohol frequency (1–2 days a month)	221.4/722 28%	62.5/298 21%	121/356 34%	0.001
Alcohol frequency (3–9 days a month)	348/722 44%	137/298 46%	150/356 42%	
Alcohol frequency (10–19 days a month)	150/722 19%	65.5/298 22%	60.5/356 17%	
Alcohol frequency (>20 days a month)	71/722 9%	33/298 11%	249/356 7%	

**Table 4 healthcare-13-00185-t004:** Characteristics related to THC consumption.

	Total	Men 194	Women 247	*p*
Cannabis smokers	441/802 55%	116/194 60%	126/247 51%	0.015
THC smoke time (<5 years)	159/441 36%	30/116 26%	60/126 48%	0.001
THC smoke time (5–10 years)	216/441 49%	67/116 58%	49/126 39%	
THC smoke time (>10 anni)	66/441 15%	18.5/116 16%	16/126 13%	
THC units	1 (0/2); 2 ± 5	1 (0/2)	0.5 (0/1)	0.001
THC frequency (1–2 days)	159/441 36%	30/116 26%	67/126 53%	0.001
THC frequency (3–9 days)	94.5/441 23%	27/116 23%	29/126 23%	
THC frequency (10–19 days)	75/441 17%	21/116 18%	16/126 13%	
THC frequency (>20 days)	110/441 25%	38/116 33%	14/126 11%	

**Table 5 healthcare-13-00185-t005:** Characteristics related to physical activity.

	Total	Men 352	Women 450	*p*
MET	2100 (818/4080)	2500 (938/5182)	1650 (720/3400)	0.001
Inactive (MET < 700)	128/802 16%	46/352 13%	81/450 18%	0.001
Moderately active (MET 700–2519)	321/802 40%	116/352 33%	193.5/450 43%	
Active and very active (MET > 2520)	353/802 44%	190/352 54%	175.5/450 39%	

**Table 6 healthcare-13-00185-t006:** Diet-related characteristics.

	Total	Men 352	Women 450	*p*
Mini ECCA score	8 (6/9)	7 (6/9)	8 (7/10)	0.001
Healthy diet	176/802 22%	53/352 15%	121.5/450 27%	0.001
Moderately healthy diet	431/802 49%	162/352 46%	229.5/450 51%	
Poor diet	200.5/802 25%	116/352 33%	90/450 20%	
Health at risk due to diet	240.6/802 3%	18/352 5%	9/450 2%	

**Table 7 healthcare-13-00185-t007:** Differences between subjects with and without urinary symptoms.

	IPSS < 3	IPSS ≥ 3	*p*
Age	26 (24/28)	25 (23/28)	0.112
Male	108/234 (46%)	250/568 (44%)	0.090
Female	126/234 (54%)	318/568 (56%)	

**Table 8 healthcare-13-00185-t008:** Differences between subjects with and without urinary symptoms in terms of cigarette smoking.

Cigarette Smoking Analysis	IPSS < 3 234	IPSS ≥ 3 568	*p*
Smokers	142/234 (61%)	431/568 (76%)	0.002
Tobacco units (number of cigarettes)	4 (2/10)	5 (1/10)	0.729
Smoking time (<5 years)	15.6/142 (11%)	80/431 (18.6%)	0.343
Smoking time (5–10 years)	33.5/142 (23.6%)	11.6/431 (27%)	
Smoking time (>10 years)	15/142 (10.4)	56/431 (13%)	
Smoking frequency (1–2 days)	6.5/142 (4.6%)	39/431 (9%)	0.121
Smoking frequency (3–9 days)	11/142 (8%)	28/431 (6.5%)	
Smoking frequency (10–19 days)	8.5/142 (6%)	20/431 (4.6%)	
Smoking frequency (>20 days)	38/142 (27%)	129/431 (30%)	

**Table 9 healthcare-13-00185-t009:** Differences between subjects with and without urinary symptoms in terms of alcohol consumption.

Alcohol Consumption Analysis	IPSS < 3 234	IPSS ≥ 3 568	*p*
Alcohol consumption	201/234 (86%)	540/568 (95%)	0.241
Alcohol units	2 (1/4)	2 (1/4)	0.031
Alcohol time (<5 years)	36/201 (18%)	108/540 (20%)	0.940
Alcohol time (5–10 years)	94/201 (47%)	238/540 (44%)	
Alcohol time (>10 years)	71/201 (35%)	194/540 (36%)	
Alcohol frequency (1–2 days)	50/201 (25%)	162/540 (30%)	0.232
Alcohol frequency (3–9 days)	97/201 (49%)	226/540 (42%)	
Alcohol frequency (10–19 days)	40/201 (20%)	103/540 (19%)	
Alcohol frequency (>20 days)	14/201 (7%)	49/540 (9%)	

**Table 10 healthcare-13-00185-t010:** Differences between subjects with and without urinary symptoms in terms of THC consumption.

	IPSS < 3 234	IPSS ≥ 3 568	*p*
Cannabis smokers	127/234 (49%)	335/568 (58%)	0.126
THC units	1 (0/2)	1 (0/2)	0.953
THC smoking time (<5 years)	36/127 (28%)	134/335 (40%)	0.193
THC smoking time (5–10 years)	67/127 (53%)	161/335 (48%)	
THC smoking time (>10 years)	23/127 (18%)	44/335 (13%)	
THC frequency (1–2 days)	47/127 (37%)	121/335 (36%)	0.234
THC frequency (3–9 days)	27/127 (21%)	80/335 (24%)	
THC frequency (10–19 days)	13/127 (10%)	64/335 (19%)	
THC frequency (>20 days)	42/127 (33%)	74/335 (22%)	

**Table 11 healthcare-13-00185-t011:** Differences between subjects with and without urinary symptoms in terms of physical activity.

	IPSS < 3	IPSS ≥ 3	*p*
MET	234	568	0.204
Inactive	35/234: 15%	91/568: 16%	0.953
Moderately active	103/234: 44%	210/568: 37%	
Active and very active	96/234: 41%	267/568: 47%	

**Table 12 healthcare-13-00185-t012:** Differences between subjects with and without urinary symptoms in terms of diet.

	IPSS < 3 234	IPSS ≥ 3 568	*p*
1.5 L of water	222/234: 95%	540/568: 95%	1.00
At least 200 g of fruits	208/234: 89%	477/568: 84%	0.128
At least 100 g of fish	215/234: 92%	505.5/568: 89%	0.343
No sugary drinks	105/234: 45%	261/568: 46%	0.857
No processed food	42/234: 18%	125/568: 22%	0.572
No sweets	44/234: 19%	119/568: 21%	0.656
At least 300 g of legumes	187/234: 80%	426/568: 75%	0.205
No processed cereals	133/234: 57%	403/568: 71%	0.008
No meals out	70/234: 30%	227/568: 40%	0.020
At least 200 g of vegetables	199/234: 85%	466/568: 82%	0.732
No red meat	147/234: 63%	409/568: 72%	0.025
No unsaturated fats	232/234: 99%	568/568: 100%	0.308

**Table 13 healthcare-13-00185-t013:** Multivariate analysis of risk of urinary symptoms.

	Multivariate OR (95% CI)	*p*	Multivariate OR (95% CI)	*p*
Age	0.98 (0.96–1.0)	0.194	0.98 (0.96–1.0)	0.194
Male/female	1.11 (0.70–2.34)	0.141		
Alcohol consumption	1.32 (0.69–2.53)	0.389		
Alcohol time	0.96 (0.77–1.20)	0.736		
Alcohol frequency	0.99 (0.83–1.18)	0.940		
Alcohol units	1.03 (1.00–1.07)	0.027		
Tobacco smokers	1.68 (1.19–2.36)	0.003	1.78 (1.26–2.65)	0.001
Smoke time	0.86 (0.64–1.16)	0.340		
Smoke frequency	0.91 (0.76/1.10)	0.379		
Tobacco units (number of cigarettes)	1.02 (0.98/1.05)	0.264		
Cannabis smokers	1.21 (0.89/1.64)	0.221		
THC smoking time	0.71 (0.48/1.04)	0.079		
THC frequency	0.91 (0.70/1.19)	0.528		
THC Units	1.07 (0.97/1.18)	0.144		
IIEF	0.99 (0.75/1.31)	0.974		
MET	1 (1/1)	0.320		
Physical activity	REF (Inactive)			
Moderate	0.80 (0.46/1.38)	0.801		
Active and very active	1.08 (0.63/1.86)	0.780		
Mini ECCA	1.12 (0.98/1.22)	0.233		

## Data Availability

The data presented in this study are available on request from the corresponding author.

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
