# Peer review of "The Impact of a Healthy Lifestyle on Lower Urinary Tract Symptoms and Erectile Function: A Prospective Study"

_healthcare, 2025, doi:10.3390/healthcare13020185_

Round 1

Reviewer 1 Report

Comments and Suggestions for Authors

Suggestions for improvement:

Introduction:

- Expand the introduction with additional references discussing LUTS in young adults to strengthen the background

- Clarify why this demographic (<30 years) is critical to study compared to older populations.

Methods:

- Provide more details on the rationale for excluding participants with specific comorbidities. This will help readers to understand the robustness of their sample selection.

- Explain how the sample size was determined and whether it is sufficient to detect smaller but potentially clinically relevant associations.

Results:

- While the results are clearly presented, add graphical representations (e.g., bar charts and scatter plots) to visualize key findings, particularly the significant associations between smoking and LUTS.

- Include more discussion on the null findings for alcohol, diet, and physical activity to explore potential reasons for the lack of significance.

Discussion:

Delve deeper into the potential mechanisms linking smoking to LUTS in young adults. highlights whether these findings align with the existing theories or provide new insights.

- Addressing the implications of the null results more thoroughly. Could other unmeasured variables (e.g., stress and sleep patterns) explain these outcomes?

Limitations:

- The study's limitations are well noted, but we consider discussing the potential effects of regional or cultural factors on lifestyle habits and LUTS prevalence.

Acknowledge the cross-sectional design as a limitation in establishing causality and suggest directions for longitudinal follow-up studies.

Practical implications:

- Frame your conclusions to emphasize actionable steps, such as public health campaigns targeting smoking cessation among young adults, to reduce LUTS prevalence.

Highlight the importance of further research to explore the role of other modifiable factors such as diet and physical activity.

Language and Presentation:

Addressing minor grammatical and stylistic errors for smoother readability.

Ensure consistent use of abbreviations and technical terms throughout the manuscript.

By addressing these comments, your manuscript can achieve greater clarity, scientific rigor, and reader engagement. Thank you for contributing this valuable research to the field.

Comments on the Quality of English Language

The quality of the English in the manuscript is generally good, with clear and coherent communication of ideas. However, there are areas in which improvements can enhance readability and professional tone. The following are some specific observations and suggestions.

Areas for improvement:

There are a few minor grammatical issues, such as inconsistent verb tense and misplaced modifiers. Example: "Alcohol consumers did present urinary symptoms more frequently..." could be revised for smoother readability as "Alcohol consumers presented urinary symptoms more frequently..."

-Some phrases are unnecessarily complex or repetitive. Simplifying the language can improve readability. For instance, "Diet, physical activity and alcohol intake were not statistically significant in young adults..." could be revised to "Diet, physical activity, and alcohol intake showed no statistically significant effects in young adults..."

-Long sentences can be broken into shorter ones for better comprehension. Example: "Despite the large sample size, the low incidence of erectile dysfunction and serious urinary symptoms limits the analysis; however, a study on a larger scale is ongoing and results will be soon available." This can be revised into two sentences for clarity.

-Ensure consistent use of technical terms and abbreviations. For example, abbreviations such as LUTS and IPSS are sometimes overused or introduced without sufficient explanation for non-specialist readers.

-Avoid informal phrases, such as "it is reasonable to promote." Instead, use formal alternatives like "promoting is warranted."

-A few typographical errors and spacing issues were present. Example: "No unsatured fats" should be "No unsaturated fats."

Author Response

We thank the reviewer for taking time to revise the manuscript. Every criticism has been deeply evaluated.

Introduction:

- Expand the introduction with additional references discussing LUTS in young adults to strengthen the background

We thank the reviewer for his/her comments and for the possibility to improve out manuscript.

Introduction section has been expanded as follows:

About half of young adults (age 39 and younger) may be affected by a variety of conditions that can cause LUTS. Filling phase symptoms are almost two-fold more common than emptying phase symptoms. Infectious or inflammatory causes are the most frequent cause of LUTS in this age group, followed by LUTS due to other etiologies that include urethral strictures, primary bladder neck obstruction, as well as neurogenic and non-neurogenic bladder dysfunction (7). As well in young adults, the presence of a bladder neck stricture is a common condition often underestimated. Its incidence in male patients with LUTS between 18-50 years old is 28-54% according to epidemiological studies. These patients usually present low body mass index, low PSA levels, severe urinary symptoms and poor flow. Recently, the pathophysiology of this condition has been challenged moving from a congenital condition to an acquired inflammatory condition. More specifically an initial infection causes a chronic inflammatory condition which then leads to a bladder neck sclerosis. Future studies should better assess the etiology of this condition in young adults. Regarding neurological conditions, several congenital and acquired conditions may cause a neurogenic bladder leading to different urodynamic profile which are not within the scope of this article. 

See New references:

Sussman RD, Drain ABB. Primary bladder neck obstruction. Rev Urol. 2019;21:53–62.

Kaplan SA, Te AE, Jacobs BZ. Urodynamic evidence of vesical neck obstruction in men with misdiagnosed chronic nonbacterial prostatitis and the therapeutic role of endoscopic incision of the bladder neck. J Urol. 1994;1521:2063–5.

Schifano N, Capogrosso P, Matloob R, Boeri L, Candela L, Fallara G, et al. Patients presenting with lower urinary tract symptoms who most deserve to be investigated for primary bladder neck obstruction. Sci Rep. 2021;11:4167

Cash, H., Wendler, J.J., Minore, A. et al. Primary bladder neck obstruction in men—new perspectives in physiopathology. Prostate Cancer Prostatic Dis 27, 54–57 (2024). https://doi.org/10.1038/s41391-023-00691-1

- Clarify why this demographic (<30 years) is critical to study compared to older populations.

We thank the reviewer for his/her comments. Most of the evidence on LUTS is on older patients why the population under 30 years old is poorly studied. This concept is now better explained in the introduction section.

‘In the past years, several studies have evaluated the role of lifestyle habits and physical activity in patients with LUTS and BPH with conflicting results. Most of these studies include male patients over 50 years old leaving apart younger patients. Therefore, very little is known about the association between LUTS and lifestyle in the young population below 30 years old.’

Methods:

- Provide more details on the rationale for excluding participants with specific comorbidities. This will help readers to understand the robustness of their sample selection.

We thank the reviewer for his/he suggestion. Methods section has been updated following the reviewers comments.

See Methods section:

The decision to exclude patients with comorbidities was made to homogenize the population enrolled and exclude possible biases in the analysis.

- Explain how the sample size was determined and whether it is sufficient to detect smaller but potentially clinically relevant associations.

We thank the reviewer for his/her comment. Sample size is now detailed in the methods section.

See Methods section:

The sample size was calculated using statistical software, estimating an 8% prevalence of LUTS in this population. The hypothesis was that individuals with LUTS exhibit poor lifestyle choices twice as frequently as those without LUTS. We used a 95% confidence interval and 90% power. The minimum sample size required was 680 patients. Considering a 20% ineligible rate, the adjusted sample size was 816.

See Results section:

Overall, 816 young adults were enrolled and 802 were eligible for analysis.

Results:

- While the results are clearly presented, add graphical representations (e.g., bar charts and scatter plots) to visualize key findings, particularly the significant associations between smoking and LUTS.

We thank the reviewer for his/her comment. A Graphic has been added as suggested.

Se New figure 1

- Include more discussion on the null findings for alcohol, diet, and physical activity to explore potential reasons for the lack of significance.

‘We thank the reviewer for his/her comment. Discussion has been improved as suggested:

In the present study no clear association was observed between alcohol, diet, physical activity and LUTS defined as IPSS≥3. The negative results clearly depend on the enrolled population. Overall, most of the enrolled population (>80%) presented a moderate consumption of alcohol, performed a moderate to active physical activity and only 3%had a diet at risk of health. As well, the present study did not evaluate other possible factors that may influence LUTS such as sleeping patterns, stress, workload and working shifts which may have an influence on LUTS . Recently, Chen et al found in their cross-sectional study that sleeping less than 6 h a night increases the risk of LUTS (OR:1,38; p<0,05). The present study confirms that in a young population slight variations in life-style do not have an impact on urinary symptoms. Future studies in different populations such as young obese, young alcoholic or severely sedentary young adults may observe different results.

Discussion:

Delve deeper into the potential mechanisms linking smoking to LUTS in young adults. highlights whether these findings align with the existing theories or provide new insights.

We thank the reviewer for his/her comments. Discussion section includes the mechanisms linking smoking to LUTS in young adults.

See Discussion section:

A central and fundamental role is played by inflammation. A sedentary lifestyle, poor physical activity, cigarette smoking, alcohol abuse, depression, hypertension, cardiovascular diseases, hyperlipidemia, diabetes mellitus, obesity, high BMI, hypogonadism are all potential factors triggering prostate and/or bladder inflammation.

De Nunzio and collaborators (3)analyzed these possibilities in a systematic review pathophysiological mechanism. The cells of the inflammatory infiltrate such as lymphocytes and macrophages are activated by numerous growth factors such as IL8, FGF-2, FGF-7 and TGF-beta. Any type of stimulus triggers the activation of a cascade mechanism resulting in hypoxia and the production of free radicals and reactive oxygen species (ROS). The picture therefore results in cellular damage and consequent abnormal reparative tissue proliferation of the cells under the stimulus of these growth factors (Figure 1). Similarly, in the bladder, reduced oxygenation/ischemia at the pelvic level can lead to an alteration of the urothelium and bladder afferent endings, triggering the inflammatory cascade (3).

In recent years, numerous studies have been published on the association between tobacco smoking and LUTS with conflicting results (24).

Data regarding nocturia associated with smoking as a risk factor for LUTS were conflicting in both male and female subjects depending on the samples examined. In one study, no particular associations were found between smoking and LUTS (p<0.05) (27). On the contrary, other studies suggest a correlation between smoking and increased frequency and urgency of urination in women, while in men no association between LUTS and tobacco has been found. In particular, consumption of 16 or more cigarettes per day compared to 0 cigarettes per day was evaluated in the Asplund study (OR, 1.8; CI, 1.1–2.8; p<0.05) and in Tettamanti's study (OR 1.5, 95% CI 1.18- 1.89) (28,29).

In the systematic review by Rohrmann S., no study demonstrated a statistically significant association between cigarette smoking and LUTS (OR= 2.01; 95% CI1.04-3.89; p<0.05) (11). While in the Platz study a significant association appears to be between smoking and LUTS, in fact heavy smokers reported a greater risk of developing urinary symptoms compared to non-smokers (OR=1.45, 95% CI 1.07-1.97; p <0.05), and moderate smokers do not (25). Our data finds a positive comparison with the Platz E study, confirming an almost double risk of developing LUTS in habitual smokers compared to non-smokers. The present study adds further evidence on the role of cigarette smoking particularly in young adults.

- Addressing the implications of the null results more thoroughly. Could other unmeasured variables (e.g., stress and sleep patterns) explain these outcomes?

We thank the reviewer for his/her comments and for the possibility to improve our manuscript. Null results are now better explained in the discussion section. See Discussion:

In the present study no clear association was observed between alcohol, diet, physical activity and LUTS defined as IPSS≥3. The negative results clearly depend on the enrolled population. Overall, most of the enrolled population (>80%) presented a moderate consumption of alcohol, performed a moderate to active physical activity and only 3%had a diet at risk of health. As well, the present study did not evaluate other possible factors that may influence LUTS such as sleeping patterns, stress, workload and working shifts which may have an influence on LUTS. Recently, Chen et al found in their cross sectional study that sleeping less than 6 h a night increases the risk of LUTS (OR:1,38 ;p<0,05). The present study confirms that in a young population slight variation in lifestyle do not have an impact on urinary symptoms. Future studies in different populations such as young obese, young alcoholic or severely sedentary young adults may observe different results. ‘

Limitations:

- The study's limitations are well noted, but we consider discussing the potential effects of regional or cultural factors on lifestyle habits and LUTS prevalence. Acknowledge the cross-sectional design as a limitation in establishing causality and suggest directions for longitudinal follow-up studies.

We thank the reviewer for his/her comments.

The limitations have been added as suggested particularly regarding the design of the study.  

See Limitation section:

‘Our study has some limitations. Our analysis applies to a series of subjects under 30 years of age and cannot be extended to other populations with different characteristics, lifestyle habits and regional or cultural factors. Despite the large sample size, the low incidence of erectile dysfunction and serious urinary symptoms limits the analysis. Anyway, a study on a larger scale is ongoing and results will be soon available. Finally, our study evaluated lifestyles at a specific point in time and could not evaluate changes in lifestyles over time. In this regard, a prospective study is underway to evaluate the effect of lifestyle changes on urinary symptoms. Finally, the present study presents the common characteristics of a cross-sectional study and therefore studies with different design and longer follow-up are ongoing. Despite the limitations, this represents one of the few papers present in the literature on the relationship between LUTS and lifestyles in the young population (<30 years old).’

Practical implications:

- Frame your conclusions to emphasize actionable steps, such as public health campaigns targeting smoking cessation among young adults, to reduce LUTS prevalence. Highlight the importance of further research to explore the role of other modifiable factors such as diet and physical activity.

We thank the reviewer for his/her comments. Conclusions have been improved as follows.

In conclusion the present study suggests a correlation between LUTS and different lifestyles in the young population. In particular, cigarette smoking was the most represented risk factor for urinary symptoms in the under 30 population, with an almost double risk for habitual smokers compared to non-smokers. The role of cannabis in the clinical picture of LUTS still remains to be defined. Diet, physical activity and alcohol intake showed no statistically significant in young adults. Anyway, given the extensive literature on samples of adult subjects, a healthy diet with reduced alcohol intake and moderate to high physical activity should be recommended. These data also highlight how prevention and awareness campaigns on certain voluptuary habits and on quitting smoking can have a significant impact on urinary health even among the young population. The present study presents some limitations related to the cross-sectional design and characteristics of the enrolled population.  Future studies, with different designs and longer follow-up, may improve our knowledge on the complex association between lifestyles and LUTS. Particularly the impact of lifestyle changing on LUTS should be evaluated. 

Language and Presentation:

Addressing minor grammatical and stylistic errors for smoother readability.

Ensure consistent use of abbreviations and technical terms throughout the manuscript.

By addressing these comments, your manuscript can achieve greater clarity, scientific rigor, and reader engagement. Thank you for contributing this valuable research to the field.

Comments on the Quality of English Language

The quality of the English in the manuscript is generally good, with clear and coherent communication of ideas. However, there are areas in which improvements can enhance readability and professional tone. The following are some specific observations and suggestions.

Areas for improvement:

There are a few minor grammatical issues, such as inconsistent verb tense and misplaced modifiers. Example: "Alcohol consumers did present urinary symptoms more frequently..." could be revised for smoother readability as "Alcohol consumers presented urinary symptoms more frequently..."

-Some phrases are unnecessarily complex or repetitive. Simplifying the language can improve readability. For instance, "Diet, physical activity and alcohol intake were not statistically significant in young adults..." could be revised to "Diet, physical activity, and alcohol intake showed no statistically significant effects in young adults..."

-Long sentences can be broken into shorter ones for better comprehension. Example: "Despite the large sample size, the low incidence of erectile dysfunction and serious urinary symptoms limits the analysis; however, a study on a larger scale is ongoing and results will be soon available." This can be revised into two sentences for clarity.

-Ensure consistent use of technical terms and abbreviations. For example, abbreviations such as LUTS and IPSS are sometimes overused or introduced without sufficient explanation for non-specialist readers.

-Avoid informal phrases, such as "it is reasonable to promote." Instead, use formal alternatives like "promoting is warranted."

-A few typographical errors and spacing issues were present. Example: "No unsatured fats" should be "No uns

We thank the reviewer for his/her suggestions. Accordingly, the manuscript has been reviewed by a native English speaker to improve readability.  

Chess-Williams R, McDermott C, Sellers DJ, West EG, Mills KA. Chronic

psychological stress and lower urinary tract symptoms. Low Urin Tract Symptoms.

2021 Oct;13(4):414-424. doi: 10.1111/luts.12395. Epub 2021 Jun 16. PMID:

34132480.

Chen J, Liu Z, Yang L, Zhou J, Ma K, Peng Z, Dong Q. Sleep-related disorders

and lower urinary tract symptoms in middle-aged and elderly males: a cross-

sectional study based on NHANES 2005-2008. Sleep Breath. 2024 Mar;28(1):359-370.

doi: 10.1007/s11325-023-02927-9. Epub 2023 Sep 29. PMID: 37775620.

DE Nunzio C, Nacchia A, Cicione A, Sica A, Baldassarri V, Voglino O, Mancini

E, Guarnotta G, Trucchi A, Tubaro A. Night shift workers refer higher urinary

symptoms with an impairment quality of life: a single cohort study. Minerva Urol

Nephrol. 2021 Dec;73(6):831-835. doi: 10.23736/S2724-6051.20.03735-2. Epub 2020

Apr 10. PMID: 32284530.

Li Marzi V, Musco S, Lombardo R, Cicione A, Gemma L, Morselli S, Gallo ML,

Serni S, Campi R, De Nunzio C; Italian Society of Urodynamics (SIUD) Young

Research Group. Prevalence of lower urinary tract symptoms in taxi drivers: a

cross-sectional web-based survey. Prostate Cancer Prostatic Dis. 2024

Jun;27(2):283-287. doi: 10.1038/s41391-023-00777-w. Epub 2023 Dec 30. PMID:

Reviewer 2 Report

Comments and Suggestions for Authors

This manuscript addresses a clinically relevant topic related to lower urinary tract symptoms (LUTS) in the adult population. While the results are interesting, the article presents several areas that need to be revised to improve clarity and scientific rigor.

The introduction mentions risk factors such as physical activity and alcohol consumption, but more scientific evidence should be included to support these points. It would be beneficial to integrate recent studies exploring how regular physical activity may influence LUTS symptoms, especially in comparison to previous studies showing contradictory results. Additionally, it would be helpful to explain in more detail the biological or physiological relationship that may link these factors to urinary symptoms.

The methodology presents a good structure, but it is essential to include explicit approval from an ethics committee. This is a crucial aspect, especially in studies involving health data.

The discussion currently does not effectively contrast the results with the existing literature. A deeper comparison with recent studies analyzing the impact of physical activity and alcohol on urinary health would be useful. Furthermore, the discussion includes information that would be more appropriate in the introduction, such as background and context, which should be reorganized to ensure the discussion focuses on interpreting the results and their relationship to other studies. It is recommended to remove personal opinions that do not align with the data presented.

In the conclusion, the use of the first person ("we") should be avoided, and a more objective, academic tone should be adopted. The conclusion should focus on the main findings, their clinical impact, and implications for future research. Additionally, it would be helpful to suggest specific areas of future research based on the obtained results.

Reviewing the citation format to comply with editorial guidelines is essential. Some references do not follow the correct style (for example, the omission of italics in journal names). A more thorough review of the references will ensure that the manuscript meets the journal's standards.

Author Response

We thank the reviewer for taking time to review our manuscript. Every criticism has been carefully aswered. 

This manuscript addresses a clinically relevant topic related to lower urinary tract symptoms (LUTS) in the adult population. While the results are interesting, the article presents several areas that need to be revised to improve clarity and scientific rigor.

We thank the reviewer for his/her comments and for the possibility to improve our manuscript.

The introduction mentions risk factors such as physical activity and alcohol consumption, but more scientific evidence should be included to support these points. It would be beneficial to integrate recent studies exploring how regular physical activity may influence LUTS symptoms, especially in comparison to previous studies showing contradictory results. Additionally, it would be helpful to explain in more detail the biological or physiological relationship that may link these factors to urinary symptoms.

We thank the reviewer for his/her comments. Introduction section has been improved as suggested:

See Introduction section:

In particular, low levels of physical activity and a high body mass index (BMI) are well-documented risk factors for LUTS in both men and women (10). The study conducted by Raheem O. and Parsons J. highlights that regular, medium-to-high intensity physical exercise can significantly lower the risks of BPH and LUTS. Both studies found that engaging in moderate to vigorous physical activity was linked to up to a 25% reduction in the risk of BPH or LUTS, with the protective benefits increasing as activity levels rose.

Data on alcohol is somehow controversial. Moderate alcohol intake appears to be protective against BPH, but the same effect does not appear to apply to LUTS in both sexes (9). Several systematic reviews have examined the link between alcohol consumption and urinary symptoms, often with conflicting results. Rohrmann S. found that consuming ≥38 g/day of alcohol reduced the risk of LUTS (OR = 0.41; p<0.05), while small daily doses also lowered risk (OR = 0.59; p<0.05). Similarly, reviews by Bradley C. and Platz E. reported moderate alcohol intake (30.1–50 g/day) was associated with a reduced risk of BPH and LUTS (OR = 0.59; p<0.05), though the protective effect diminished at higher levels (≥50.1 g/day, OR = 0.72; p<0.05).

The methodology presents a good structure, but it is essential to include explicit approval from an ethics committee. This is a crucial aspect, especially in studies involving health data.

We thank the reviewer for his/her comments and for the possibility to improve our manuscript. Data on Ethics Committee has been added.

See Methods section:

This study was evaluated and approved by the Bioethical Committee of our University. Prot. 258 SA_2021 RIF.  CE  6376_2021 -  Studio Clinico:“IRU STUDY’

The discussion currently does not effectively contrast the results with the existing literature. A deeper comparison with recent studies analyzing the impact of physical activity and alcohol on urinary health would be useful. Furthermore, the discussion includes information that would be more appropriate in the introduction, such as background and context, which should be reorganized to ensure the discussion focuses on interpreting the results and their relationship to other studies. It is recommended to remove personal opinions that do not align with the data presented.

We thank the reviewer for his/her comments and for the possibility to improve our manuscript. A deeper comparison with recent studies is now described in the discussion section and the background moved to the introduction section.

See new discussion section.  

In the conclusion, the use of the first person ("we") should be avoided, and a more objective, academic tone should be adopted. The conclusion should focus on the main findings, their clinical impact, and implications for future research. Additionally, it would be helpful to suggest specific areas of future research based on the obtained results.

We thank the reviewer for his/her comments and for the possibility to improve our manuscript. The conclusion section has been improved as suggested by the reviewer. See New Conclusion section:

In conclusion the present study suggests a correlation between LUTS and different lifestyles in the young population. In particular, cigarette smoking was the most represented risk factor for urinary symptoms in the under 30 population, with an almost double risk for habitual smokers compared to non-smokers. The role of cannabis in the clinical picture of LUTS still remains to be defined. Diet, physical activity and alcohol intake showed no statistically significant in young adults. Anyway, given the extensive literature on samples of adult subjects, a healthy diet with reduced alcohol intake and moderate to high physical activity should be recommended. These data also highlight how prevention and awareness campaigns on certain voluptuary habits and on quitting smoking can have a significant impact on urinary health even among the young population. The present study presents some limitations related to the cross-sectional design and characteristics of the enrolled population.  Future studies, with different designs and longer follow-up, may improve our knowledge on the complex association between lifestyles and LUTS. Particularly the impact of lifestyle changing on LUTS should be evaluated. 

Reviewing the citation format to comply with editorial guidelines is essential. Some references do not follow the correct style (for example, the omission of italics in journal names). A more thorough review of the references will ensure that the manuscript meets the journal's standards.

We thank the reviewer for his/her comments and for the possibility to improve our manuscript. References have been carefully reviewed.

Reviewer 3 Report

Comments and Suggestions for Authors

Dear Authors,

I am pleased to have had the opportunity to review your manuscript entitled: “The impact of a healthy lifestyle on LUTS and erectile function: a prospective study” Below are a number of changes that I believe need to be reviewed prior to publication:

- The title of the manuscript should not carry acronyms to make it easier for readers.

- The abstract starts directly with the objective of the study. Try to adapt the abstract to IMRyD format.

- In the introduction there are paragraphs that are not related to any citation such as the following example “LUTS can be distinguished in voiding, storage and post micturition symptoms and are often considered a consequence of benign prostatic hyperplasia (BPH). BPH is a common cause of bladder outlet obstruction (BOO) with an increasing incidence in men over 50 yr of age. Bladder dysfunctions may also cause BOO, like detrusor overactivity (OAB) and detrusor underactivity (DU); other dysfunctions of the urinary tract can be cause of LUTS.” Verify that each idea is cited.

- At the end of the introduction it would be interesting to add a subsection referring to the objective and hypotheses of the study.

- Section 2. Materials and Methods should be divided into different subsections: Participants and data collection (where sociodemographic data of the participants such as age and gender are presented), Instruments (indicate number of items, example of an item and reliability obtained), Data analysis (statistical tests used and what each test was used for). It is also important to add in this section the approval obtained by the Bioethics Committee for the study.

- Relate the discussion to the initial hypotheses (which I have suggested should be added as an additional section in the Introduction).

- Include the limitations of the study and future proposals in the Conclusions section.

I wish you success in your research,

Best regards. 

Author Response

We thank the reviewer for taking time to review our manuscript every criticism has been carefully evaluated. 

Dear Authors,

I am pleased to have had the opportunity to review your manuscript entitled: “The impact of a healthy lifestyle on LUTS and erectile function: a prospective study” Below are a number of changes that I believe need to be reviewed prior to publication:

- The title of the manuscript should not carry acronyms to make it easier for readers.

We thank the reviewer for his/her comments and for the possibility to improve our manuscript. Title has been modified as follows: ‘The impact of a healthy lifestyle on Lower Urinary Tract Symptoms and erectile function: a prospective study’

- The abstract starts directly with the objective of the study. Try to adapt the abstract to IMRyD format.

We thank the reviewer for his/her comment we have adapted the abstract according to IMRyD format.

- In the introduction there are paragraphs that are not related to any citation such as the following example “LUTS can be distinguished in voiding, storage and post micturition symptoms and are often considered a consequence of benign prostatic hyperplasia (BPH). BPH is a common cause of bladder outlet obstruction (BOO) with an increasing incidence in men over 50 yr of age. Bladder dysfunctions may also cause BOO, like detrusor overactivity (OAB) and detrusor underactivity (DU); other dysfunctions of the urinary tract can be cause of LUTS.” Verify that each idea is cited.

We thank the reviewer for his/her comments. Citations have been updated as suggested.

- At the end of the introduction it would be interesting to add a subsection referring to the objective and hypotheses of the study.

We thank the reviewer for the suggestion. A subsection has been added as follows:

1.1 Objective and Hypothesis

The aim of the present study is to evaluate the association between diet, physical activity, smoking, alcohol consumption and LUTS in young adults. He hypothesis is thar an adequate lifestyle is associated with a lower risk of LUTS in this population of young adults.

- Section 2. Materials and Methods should be divided into different subsections: Participants and data collection (where sociodemographic data of the participants such as age and gender are presented), Instruments (indicate number of items, example of an item and reliability obtained), Data analysis (statistical tests used and what each test was used for). It is also important to add in this section the approval obtained by the Bioethics Committee for the study.

We thank the reviewer for the suggestion. The revisions suggested are now present in the new version of the manuscript.

- Relate the discussion to the initial hypotheses (which I have suggested should be added as an additional section in the Introduction).

We thank the reviewer for his/her comments. Discussion is now aligned to the hypothesis discussing the available literature on lifestyle and LUTS.

- Include the limitations of the study and future proposals in the Conclusions section.

We thank the reviewer for his/her comments. The Conclusion section has been improved as suggested by the reviewer. See new Conclusion section:

In conclusion the present study suggests a correlation between LUTS and different lifestyles in the young population. In particular, cigarette smoking was the most represented risk factor for urinary symptoms in the under 30 population, with an almost double risk for habitual smokers compared to non-smokers. The role of cannabis in the clinical picture of LUTS still remains to be defined. Diet, physical activity and alcohol intake showed no statistically significant in young adults. Anyway, given the extensive literature on samples of adult subjects, a healthy diet with reduced alcohol intake and moderate to high physical activity should be recommended. These data also highlight how prevention and awareness campaigns on certain voluptuary habits and on quitting smoking can have a significant impact on urinary health even among the young population. The present study presents some limitations related to the cross-sectional design and characteristics of the enrolled population.  Future studies, with different designs and longer follow-up, may improve our knowledge on the complex association between lifestyles and LUTS. Particularly the impact of lifestyle changing on LUTS should be evaluated.